# Exploring the non-linear association of daily cigarette consumption behavior and food security- An application of CMP GAM regression

**Shafeel Umam** [1,2]*, **Rubaiya Binte Razzak**[1,2], **Munniara Yesmin Munni**[3], **Azizur Rahman**[1]

**1** Department of Statistics and Data Science, Jahangirnagar University, Savar, Dhaka, **2** Department of Behavioral Science and Health Equity, College for Public Health and Social Justice, Saint Louis University, Missouri, **3** PME department, Christian Commission for Development in Bangladesh

☯ These authors contributed equally to this work.
* shafeel.umam@outlook.com

## Abstract

### Background

Food insecurity and cigarette smoking are significant public health issues that disproportionately affect vulnerable populations. Previous research often overlooks daily smoking patterns, focusing instead on smoking status or cessation. This study seeks to fill that gap, hypothesizing that food insecurity is positively associated with higher daily cigarette consumption.

### Methods

This study investigates the non-linear association between food security status and daily cigarette consumption using data from the 2022 National Health Interview Survey (NHIS). It introduces a novel statistical approach—the COM-Poisson Generalized Additive Model (CMP-GAM)—to capture the non-linear dynamics between these variables,

### Results

The CMP-GAM model was applied to a sample of 819 adults, controlling for key variables like age, body mass index (BMI), and mental health conditions (anxiety and depression severity). The results indicated that while food security status did not significantly affect daily cigarette consumption, the age of smoking initiation was a strong predictor, with earlier smoking onset leading to higher daily cigarette use. Subgroup analyses among participants with cardiovascular disease and diabetes also did not reveal a significant relationship between food insecurity and smoking frequency.

**Data availability statement:** The data underlying the results presented in the study are available from figshare: https://figshare.com/s/cdd56229a1c2cba5d50a.

**Funding:** The author(s) received no specific funding for this work.

**Competing interests:** The authors confirm that no known conflicts of interest are associated with this publication. Specifically, this research utilizes secondary data that has been fully anonymized and poses no direct or indirect benefit to any of the researchers involved. There have been no financial or personal relationships with other people or organizations that could inappropriately influence our work. This declaration is made to ensure full transparency and to uphold the integrity of the scientific process.

## Significance

The findings suggest that targeted interventions aimed at preventing early smoking initiation may be more effective in reducing smoking prevalence and improving public health outcomes than addressing food security alone. This study highlights the need for further research into the complex interplay between socioeconomic factors and health behaviors like smoking.

## 1 Introduction

In 2022, around 2.4 billion people (29.6 percent of the global population) were moderately to severely food insecure, among which 900 million (11.3 percent) faced severe food insecurity [1]. Food insecurity, characterized by limited or uncertain access to adequate food, impacts beyond nutrition and physical health, affecting various aspects of mental and social well-being. Individuals experiencing food insecurity are at higher risk of health problems such as diabetes, hypertension, mental illness, and increased mortality [2]. Moreover, Studies suggest that food insecurity affects mental health, with evidence suggesting associations with depression, anxiety, and other psychological distress [3,4]. Smoking, on the other hand, remains one of the leading causes of preventable diseases and deaths globally. Smoking has been associated with a range of health conditions, such as cardiovascular diseases, neoplasms, chronic respiratory diseases, respiratory infections, tuberculosis, musculoskeletal disorders, as well as diabetes and kidney diseases [5]. Over an extended period, smoking has a detrimental impact on an individual's health, diminishes productive workforce participation, and exerts influence on personal financial stability [6]. Understanding the factors that influence smoking behavior, particularly the number of cigarettes smoked per day, is essential for developing effective public health interventions. Several studies have investigated the connections between food security and smoking in an effort to identify underlying factors and preventive actions.

Previous research has shown that socioeconomic factors, including food security, are closely linked to smoking behavior. An earlier study among young adults observed that smoking prevalence is significantly higher for young adults who reported being food insecure [7]. The effect of food insecurity on smoking behavior is particularly noticeable among low-income and marginalized populations. Low-income adults experiencing food insecurity have been found to have a disproportionately high prevalence of cigarette smoking [8]. The impact of food insecurity on smoking behavior is multifaceted. Studies have shown that food insecurity is a barrier to smoking cessation [9]. Previous studies have discovered that facing food insecurity is associated with a higher likelihood of starting smoking for those who do not smoke [10]. Furthermore, Baek et al. noted that shifts from food security to insecurity were associated with almost a twofold rise in the likelihood of smoking behavior [11]. Individuals experiencing food insecurity may use smoking as a coping mechanism to manage stress and anxiety associated with their precarious food situation [12]. Canales et al. (2015) delineated that worrying about running out of food is an important indicator of

food security and has direct causal implications on one's perceptions of stress or anxiety [13]. Whereas cigarette smoking is a common way to relieve tension and stress [14], suggesting an association between the intensity of smoking and food insecurity. Individuals experiencing food insecurity may turn to smoking as a way to manage stress or suppress appetite, which can further complicate efforts to quit smoking [15]. Additionally, financial constraints can lead to prioritizing immediate relief through smoking over long-term health investments [16]. Psychological distress exacerbated by food insecurity may promote smoking for stress relief, creating a cycle that hinders cessation [7]. Moreover, a dose-response relationship has been observed between tobacco product use and levels of food insecurity, indicating that higher levels of smoking-induced financial deprivation can worsen food insecurity [6]. Nicotine intake reduces appetite and suppresses hunger [17], which increases the likelihood of smokers using smoking as a coping mechanism to reduce hunger related to food scarcity. This dynamic exacerbates health disparities and significantly burdens public health systems. Taking these into account, there is a growing recognition of the need for more comprehensive assessments to understand the dynamics between tobacco use and food insecurity [14].

Numerous statistical models have been utilized in research studies to explore the association between food insecurity and smoking behaviors. In most of the research studies logistic regression analysis was used to examine how food insecurity is associated with smoking status [7,12,18]. Longitudinal logistic and Tobit regressions analyzed the link between food security and smoking behaviors in low-income women, focusing on the longitudinal impact of food insecurity on smoking habits [19]. A prior study used multinomial logistic regression to assess and characterize the relationship between food security and tobacco product use [20]. Another study utilized multinomial logistic regression to examine the association between transitions of food insecurity with smoking cessation and shifts in the consumption pattern over two years among adults [21]. However, most studies only considered a linear association between food security and cigarette smoking behavior. Nevertheless, the non-linear nature of the association requires further investigation. To our knowledge, there is no study that has been conducted to capture the non-linear association between those variables using an appropriate count data model.

Considering a non-linear association between food security and cigarette smoking behavior is important for several reasons. The relationship between food security and smoking behavior may not be straightforward. Non-linear models can capture more complex interactions and thresholds that linear models might miss. For example, the impact of food insecurity on smoking behavior might be more pronounced at certain levels of food insecurity [7]. Food insecurity and smoking behavior can influence each other in multiple ways. For instance, food insecurity might lead to increased stress and smoking as a coping mechanism, while smoking can exacerbate financial strain, worsening food insecurity [14]. Understanding non-linear relationships can help in designing more effective policies and interventions. For example, interventions might need to be targeted differently at various levels of food insecurity to be effective. Incorporating non-linear associations enriches the literature by providing a more nuanced understanding of the dynamics between food security and smoking. It challenges the oversimplified view of linear relationships and encourages further research into the underlying mechanisms. Overall, by considering non-linear associations, researchers can uncover more detailed insights and develop more targeted strategies to address the intertwined issues of food insecurity and smoking behavior.

Poisson and Negative-Binomial regression are the most popular models for cross-sectional count data [22]. Besides these, the Conway-Maxwell-Poisson (CMP) or COM-Poison regression model introduced by Conway and Maxwell [23] is increasingly gaining popularity for complex association in count data as it can account for both under-dispersion and over-dispersion [22]. The CMP regression is an extension of the traditional Poisson and logistic regression models [24]. While modeling count data, the Poisson regression is limited with its variance assumption of equal mean and variance, and Negative-binomial regression can only model over-dispersed count data. While the CMP regression model is not limited to these assumptions and provide a flexible approach for modeling count data with varying dispersion levels [24], which can be helpful in allowing for a more accurate representation of the relationship between food security and the quantity of daily smoking. However, CMP regression is limited when dealing with complex nonlinear relationships [22].

Very recently, Chatla and Shmueli [22] introduced a model which overcome the limitation of traditional CMP-regression model and they referred this model as COM-Poisson Generalized additive model (CMP-GAM). In that study, they considered an iterative reweighted least square (IRLS) estimation framework and extended the CMP model with additive components using a penalized spline approach makes this model better at capturing complex non-linear relationships [22]. Considering the advance features of capturing non-linear association by this model, we consider the CMP-GAM to understand the association between food security and the number of cigarettes a person smokes in a day.

The association between food insecurity and smoking extends beyond individual behaviors and has broader implications for public health interventions [25]. Moreover, the impact of cigarette smoking behavior on food security needs to be investigated further due to its potential to inform targeted interventions. However, to date, the majority of research studies focus on analyzing smoking behaviors and cessation in relation to food security, but they often overlook the daily consumption patterns as a factor in this association. Therefore, this study aims to understand the non-linear association between food security and daily smoking behavior using CMP-GAM. Our primary hypothesis is that food security status is associated with the number of cigarettes smoked per day. Specifically, we hypothesize that individuals experiencing food insecurity will smoke more cigarettes daily compared to those with secure food access.

## 2 Methods

### Data and study design

This study used the cross-sectional study design using data from the 2022 National Health Interview Survey (NHIS) conducted by the CDC's National Center for Health Statistics (NCHS). Information from "Sample adult Interviews" was used for the study, where the interview participants were US adults aged 18 years and above. The NHIS is an annual household survey representing the civilian non-institutionalized US population across the 50 states and the District of Columbia. It collects data from individuals residing in households and noninstitutional group quarters, including homeless shelters, rooming houses, and group homes. Individuals temporarily living in student dormitories or temporary housing are sampled within their permanent households [26]. The survey excludes those without a fixed household address, active-duty military personnel and civilians on military bases, residents of long-term care institutions (such as nursing homes and chronic care hospitals), individuals in correctional facilities, and U.S. nationals living abroad [26]. However, civilians residing with military personnel in non-military housing remain eligible for inclusion. The NHIS employs a geographically clustered sampling design to select dwelling units, ensuring national representativeness [26]. Based on the 2010 decennial census, the sampling process partitions the United States into 1,689 geographic areas, comprising counties, county equivalents, or contiguous county groups, ensuring they remain within state boundaries [26]. The sample is structured in a way that monthly data is nationally representative [26]. Usually, interviews for the NHIS are conducted in-person modules and followed up via telephone. For over 50 years, the U.S. Census Bureau has sent interviewers to American homes to inquire about various health topics. Data collection is continuous, spanning from January to December each year, allowing for timely and comprehensive health surveillance [26]. The NHIS collects and analyzes data on various health topics to monitor the health of the US population. Survey results are instrumental for providing information and tracking health status, healthcare access, and advancement toward achieving national health objectives.

### Participants

For the NHIS 2022, more than 30,000 confidential interviews were conducted during the year with the help of field staff trained by the U.S. Census Bureau. Around 27,651 adult participants participated in the "Sample Adult Interviews" in NHIS 2022. The response rate for the "Sample adults interviews" was 47.7% [27].

The inclusion and exclusion criteria for this study were determined based on the variables of interest and their alignment with the research objectives. From the 27,651 responses, (n = 1,267) data were excluded as missing food security information. Additionally, non-smokers were excluded from the study (n = 25,565). Out of 27,651 respondents, only 819

met the inclusion criteria based on the specified variables. The final selection was guided by the study's research objectives to ensure relevance and accuracy in the analysis.

## Variables of interest

**2.1.1 Dependent variable.** The daily cigarette smoking rate is the primary outcome of this analysis. A self-reported survey item was used to identify the number of cigarettes consumed in a day. Participants were asked, 'On average, about how many cigarettes do you NOW smoke a day?' Following the responses to this question, the number of cigarettes consumed in a day is determined.

**2.1.2 Independent variable.** Household food security was measured using the U.S. Department of Agriculture's (USDA) 10-item-based food security module with a recall period of 30 days [28]. Based on the responses for the 10 items, household food security status was classified into four categories: High food security (scores 0), Marginal food security (scores 1–2), Low food security (scores 3–5), and very low food security (scores 6–10) [29].

**2.1.3 Covariates.** The study utilized various sociodemographic factors and disease status as covariates to account for their potential to confound the relationship between food security and the daily consumption of cigarettes. Demographic covariates included are sex (male/female), age, marital status (married but spouse is present, married but spouse is not present, married but spouse presence unknown, widowed, divorced, separated, never married, and living with partner). Health and disease-related factors included as covariates are the severity of depression symptoms, the severity of anxiety symptoms, experiencing high cholesterol (yes/no), the presence of hypertension (yes/no), diabetics (yes/no), cardiovascular disease (yes/no), and Body Mass Index (BMI) (Underweight, healthy weight, overweight, obese). The Generalized Anxiety Disorder (GAD-7) [30] questionnaire was used to assess the severity of anxiety symptoms, and the Public Health Questionnaire (PHQ-8) [31] was used to assess the severity of depression symptoms within the last two weeks. Behavioral factors included in this analysis were age at starting smoking (continuous variable) and average sleep hours (continuous variable).

## Statistical analysis

The CMP regression is particularly suited for count data, effectively handling overdispersion and underdispersion. Considering a count variable as the response variable, namely, the number of cigarettes smoked in a day, this study used a Conway-Maxwell-Poisson (CMP) regression extended to incorporate additive components using penalized splines. The model considers applicable parametric and non-parametric components for estimation, which are given below.

Parametric components — Respondents' food security status, sex, the severity of anxiety, the severity of depression, experiencing any cardiovascular disease, high cholesterol, and hypertension.

Non-parametric components—The non-parametric components of the model are the respondents' age, age at starting smoking, average sleep hours, and Body Mass Index (BMI).

When dealing with cross-sectional count data, the most commonly used regression models are Poisson and negative-binomial regression [22]. Taking this into account, results from this model were compared with outcomes from the Poisson and Negative binomial generalized additive models to investigate the nature of the association further.

**2.1.4 Sub-group analysis.** To investigate the association further, this study conducted two separate models to examine how various health conditions and disease status affect the relationship between daily cigarette consumption and household food security. The categorizations for the sub-groups were based on whether the respondents had diabetes or cardiovascular diseases. The model for cardiovascular disease was based on 114 observations, while the model for diabetes was based on 172 observations.

All statistical analyses were conducted using R (4.4.0).

## Model specification and estimation

**2.1.5 Generalized additive model.** As, additive models have the advantage to capture complicated relationship than mixed methods or normal models [32,33]. Generalized Additive Model (GAM) are used to incorporate nonlinear terms

which is an extension of Generalized Linear Models (GLM). Linear form of the covariates are being replaced by the smooth functions in the GAM model. A generalized additive model is a generalized linear model with a linear predictor involving a sum of smooth functions of covariates which has the formulation [34],

$$g(\mu_i) = X_i^* \theta + f_1(x_{1i}) + f_2(x_{2i}) + f_3(x_{3i}, x_{4i}) + \ldots$$

$$= X_i^* \theta + \sum_{j=1}^{p} f_j(x_{ij}); \qquad i = 1, \ldots, n$$

$\mu_i = E(z_i)$, $X_i^*$ is a row of the model matrix of any strictly parametric model components, $\theta$ is the parametric component, $f_i$ is the smooth functions of the covariates and $g(.)$ is a smooth monotonic and twice differentiable link function. To estimate smooth functions there exist multiple methods [34–36]. $\theta_i$ is the smoothing parameter that control the trade-off between fit and smoothness. Cubic spline smoother, backfitting algorithm, penalized spline approaches are widely used. Backfitting algorithm uses a regression type fitting algorithm, which serves as baseline method for estimating $f_i$. It has the flexibility to incorporate wide variety of smoothing methods for components. Convergence can be found in the paper to follow [35,36].

GAMs employ smoothing techniques to model non-linear relationships between variables. Commonly used smoothing methods include regression splines (like B-splines and P-splines), local regression (loess), and smoothing splines. Regression splines are particularly favored in practice due to their computational efficiency and representation as linear combinations of basis functions, making them well-suited for estimation and prediction. [37,38] A smoothing spline with two predictors can be used for fitting a wiggly surface. Splines can take more than two arguments, in which case wiggly hypersurfaces are modeled. Given a linear predictor with appropriate smooths, this linear predictor can be used to model Gaussian response variables, or Poisson or binomial responses. GAMs can also accommodate ordinal responses as well as multinomial responses.

The penalized splines approach is a popular method for fitting additive models. It represents each function in the model using spline-type basis expansions. To prevent overfitting, a penalty term is added to the likelihood function, which is then maximized. This maximization process is carried out using penalized iteratively re-weighted least squares (P-IRLS) [35]. In essence, the GAM is fitted by iteratively minimizing a penalized least squares problem,

$$\left\| \sqrt{w^K \left(T^{(k)} - X\beta\right)} \right\|^2 + \sum_j \eta_{lj} \beta^T S_j w, r, t.$$

$T^{(k)}$ denotes the adjusted response variable and $W^k$ denotes the weights at the $k^{\text{th}}$ iteration od the P-IRLS algorithm. $S_j$ measures the roughness of $f_i$, which are matrices of known coefficients $\beta^T S_j \beta$. The $\eta_j$ are smoothing parameters that control the trade-off between fit and smoothness and their selection can be achieved by minimizing the Generalized Cross Validation (GCV) score, AIC, or another criterion [35,39–42].

The Generalized Additive model for the CMP regression, Poisson and Negative Binomial is used to compare the results.

**2.1.6 Poisson generalized additive model.** The Poisson GLM is described as,

$$E(X_1, X_2, \ldots, X_p) = \beta_0 + \beta_1 X_1 + \beta_2 X_2 + \ldots + \beta_p X_p$$

The log functions for Poisson link is,

$$g(\mu) = \log \log (\mu) = \eta$$

The conditional expectations can be formulated as,

$$E(X_1, \ldots, X_p) = s_0 + s_1(X_1) + s_2(X_2) + \ldots + s_p(X_p)$$

$s_i(X)$ is the smoothing parameter. The GAM consist of a random component, an additive component, and a link function relating the two components. The response variable assuming to follow exponential family,

$$f_z(z; \theta; \phi) = exp\{\frac{y\theta - b(\theta)}{a(\phi)} + c(y, \phi)$$

Where $\theta$ is the natural parameter and $\phi$ is the scale parameter. The quantity is defined as,

$$\eta = s_0 + \sum_{i=1}^{p} s_i(X_i)$$

Where the function $s_i(X_i^*)$ are the smooth functions defines the additive component and the relationship between $\mu$ and $\eta$. A smoother is a tool for summarizing the trend of a response measurement Y as a function of one or more predictor measurements $X_1, \ldots, X_p$. It produces an estimate of the trend that is less variable than $Z$ itself. An important property of a smoother is its nonparametric nature. It doesn't assume a rigid form for the dependence of $Z$ on $X_1, \ldots, X_p$. In this study, we focus on only a cubic smoothing spline that can be used with GAM. A cubic smoothing spline is the solution to the following optimization problem: among all functions $\eta(x)$ with two continuous derivatives, find one that minimizes the penalized least square:

$$\sum_{i=1}^{n} (z_i - \eta(x_i))^2 + \lambda \int_a^b (\eta''((t))^2 dt$$

Where $\lambda$ is a fixed constant and $a \leq x_1 \leq \ldots \leq x_n \leq b$. The first term measures closeness to the data while the second term penalizes curvature in the function. It can be shown that there exists an explicit, unique minimizer and that minimizer is a natural cubic spline with knots at the unique values of $x_i$. The parameter $\lambda$ is the smoothing parameter. Further estimation process with AIC description illustrated in this paper [43–45].

**2.1.7 Negative binomial generalized additive model.** Negative binomial (NB) regression is the most common full-likelihood method for analyzing count data exhibiting overdispersion with respect to the Poisson distribution [46]. The log-likelihood for NB model for known K is,

$$l(\mu; k) = n\{k \, lnk - ln\Gamma(k)\} + \{\sum_{i=1}^{n} \{ln\Gamma Z_i + k) - (Z_i + k) \, ln(k + \mu)\} + d(Z, \mu)$$

To fit the NB model for $Z$ given covariates X, various link functions are possible. Methodology describe in [34] requires the known value of k. And for a known $\mu$, estimation of k reduces to an ordinary maximum likelihood problem which maximizes the log-likelihood. The structure of the algorithm is elaborately described in [47], for which further estimations aren't presented here.

**2.1.8 The CMP regression.** To model count response variable, Poisson regression is widely used [48]. Though its flexibility always in question for the usage of a single parameter in the model. Overdispersion and underdispersion is one of the common issues that can not be addressed through the Poisson model [23]. The Conway-Maxwell Poisson (COM-Poisson) have proven powerful in capturing wide range of dispersion [49]. Originated by [50] is a two-parameter generalization of the Poisson distribution that allows for different levels of dispersion. Though originated by Conway, the statistical properties are revived by [23]. As, the COM Poisson have the flexibility to capture overdispersion and underdispersion, the COM Poisson distribution is a two-parameter model, generalized by Poisson, Bernouli and Geometric distributions. [23,50] The probability mass function (p.m.f) has the form,

$$P(Z = z) = \frac{\lambda^z}{(z!)^\nu \, \zeta(\lambda, \nu)}, \; z = 0, 1, 2, \dots; \; \zeta(\lambda, \nu) = \sum_{j=0}^{\infty} \frac{\lambda^j}{(j!)^\nu}$$

Where Z is a random variable that follows CMP distribution with the parameters, $\lambda = E(Z^\nu) > 0$ is the generalization of the Poisson rate parameter and $\nu \geq 0$ is the dispersion parameter. The normalizing constant is denoted by $\zeta(\lambda, \nu)$. Three common distributions e.g., Poisson ($\nu = 1$, Geometric $\nu = 0$, $\lambda < 1$), Bernoulli ($\nu \to \infty$ with probability $\frac{\lambda}{\lambda+1}$ is included as special cases in the CMP distribution. The CMP distribution addresses both over dispersion $0 \leq \nu < 1$) and under dispersion ($\nu > 1$). The Mean and variance of the distribution is approximated as,

$$E(Z) = \frac{\partial ln \zeta(\lambda, \nu)}{\partial ln \lambda} = \lambda \frac{\delta ln \zeta(\lambda, \nu)}{\delta \lambda} \approx \lambda^{\frac{1}{\nu}} - \frac{\nu - 1}{2\nu}$$

$$Var(Z) = \frac{\partial^2 ln \zeta(\lambda, \nu)}{\partial (ln \lambda)^2} = \frac{\partial E(Z)}{\partial ln \lambda} \approx \frac{1}{\nu} \lambda^{\frac{1}{\nu}}$$

The detailed approximation and other distributional properties of the distribution can be found in the papers. [51,52]

The COM-Poisson regression was developed by the model formulation of these papers [53,54]. The regression form to describe the relationship between the explanatory and response variable can be stated with log link function

$$ln \, ln \, (\lambda) = X\beta \, ; \beta \in R^{p+1}$$

Which models the indirect association between **E(Z)** and **X**. The loglinear link,

$$ln \, ln \, (\nu) = G\gamma \, ; \gamma \in R^{q+1}$$

The loglink function is being used for reducing the varying dispersion in Poisson and logistic regression. □ and □ are the regression coefficients for the centering link function and the shape link function.

**2.1.9 CMP generalized additive model.** For the restrictive nature of the CMP distribution in modeling nonlinear relationships or time series data additive model for CMP regression is being developed in the paper [22]. CMP generalized model has the formulation,

$$ln(\lambda_i) = x_i^* \theta^* + \sum_{j=1}^{p} f_i(x_{ij})$$

$$ln(\nu_i) = u_i^* \delta^* + \sum_{j=1}^{k} m_j(u_{ij})$$

Where $i = 1, \dots, n$ where $\theta^*$ and $\delta^*$ are the parameter vectors the parametric part of $ln(\lambda_i)$ and $ln(\nu_i) f_j m_j$ are the smooth functions for the covariates $xj$ and $z_j$ and are subject to identifiability constraints. For further model estimation readers are encouraged to go through the paper [22].

**2.1.10 Model Estimation.** The revived properties of CMP distribution by [23] estimate the log-likelihood by

$$log \, log \, L_i(z_i) = z_i log \lambda_i - \nu \, log z_i! - log \zeta(\lambda_i, \nu)$$

Summing over n observations, the log-likelihood can be written as,

$$\log\log L = \sum_{i=1}^{n} z_i \log\lambda_i - \nu \sum_{i=1}^{n} \log z_i! \ - \sum_{i=1}^{n} \log\zeta(\lambda_i, \nu)$$

These equations uses the maximizing equation under constraint $\nu \geq 0$. In [23] Generalized linear model is used to obtain the maximum likelihood estimates. As v models the dispersion, to control it covariates need to be included which is demonstrated by two level of group data [55]. The log-likelihood can be described as,

$$l_i(z_i, \beta, \gamma) = z_i x_i^T \beta - \ln\ln \ (z_i!) \ \exp\exp\ \{g_i^T \gamma\} \ - \ln\zeta_i \left( \exp\exp\{x_i^T \beta\}, \exp\exp\ \{g_i^T \gamma\} \ \right)$$

Resulting in the score equations:

$$\frac{\partial l_i}{\partial \beta^T} = x_i \left( z_i - \frac{\partial \ln\zeta_i}{\partial \ln\lambda_i} \right) = x(z_i - E[z_i]),$$

$$\frac{\partial l_i}{\partial \gamma^T} = g_i \left[ \left( -\ln\ln\ (z_i!) \ - \frac{\partial \ln\ln\ \zeta_i}{\partial \nu_i} \right) \nu_i \right] = g_i \left[ (-\ln\ln\ (z_i) \ + E[\ln\ln\ (z_i!) \ ]) \right] \nu_i$$

The standardized generalization linear models can not be used to solve the equations. So, mostly gradient based methods are being used in CMP regression than IRLS. The detail formulation of gradient based mixed methods using GLM can be found in the paper, readers are encouraged to read the paper for the demonstration [24].

**2.1.10.1 Iterative reweighted least squares (IRLS) framework.** Because of the advantage of not affecting outliers and not being bound to use the information matrix, the blank in the field of CMP regression of not using the IRLS method has been filled by [22] to capture the complexity of nonlinear relationships. The estimation of CMP regression in this paper follows this method to generate the result. The framework of estimation is developed by five steps. First step includes the calculations the cumulants. Several authors have used different approaches to calculate cumulants [23,56,57]. Asymptotic approximation improved by including two lower terms [58]. To make this approximation close to the true value, $\lambda_i \geq 2$, $\nu_i < 1$ the cumulants are being calculated from the asymptotic terms and for other values p.m.f is used with bounding error [22]. In the second step, IRLS equations are being updated for parameters □ and □ which can be easily estimated with WLS method. In the following three steps proof of convergence of the two step method, practical issues and Inferences been described [22].

## Ethics statement

The Research Ethics Review Board (ERB) of the National Center for Health Statistics (NCHS) approved the content and methods of the National Health Interview Survey (NHIS) to protect the study participants [59]. Prior to taking part in the NHIS study, all participants provided their verbal consent. Potential NHIS respondents were informed about their rights regarding survey participation and were assured of the confidentiality of their responses, as well as the protection of their data [60]. All data in the publicly available dataset are fully anonymized prior to release. All authors declare they have no competing interests.

## 3 Results

The study used data from the National Health Interview Survey (NHIS) of 2022, a cross-sectional household survey of the US noninstitutionalized civilian population. Among which the adult module has been used.

Among 27,561 surveyed adults, 819 adults have been selected for our study, where the number of daily cigarette consumption has been used as the count variable for the application of CMP regression. As we have filtered the dataset with the variables of our interest, there was no missing data in our final dataset. The variables of our interest were Age, Age when started smoking, severity of depression, severity of anxiety, presence of cardiovascular disease, presence of

cholesterol, diabetics, hypertension, BMI, and marital status. For our two distinct models, we have selected two diseases, cardiovascular and diabetics. With the numerical functions of age, BMI, smoking age, and weight with the categorical co-variates of food security, gender, hypertension, diabetics, anxiety, and depression severity for the cardiovascular model/lambda model. For the nu model, the numerical functions are the same with the additional categorical variable of the presence of cholesterol. In both models, the count variable was the number of cigarettes consumed in a day.

Demographic characteristics of the selected variables been represented in Table 1.

The median age of the sample population was 61 years, with smoking initiation occurring at a median age of 17 years ($p < 0.001$ and $p = 0.003$, respectively). Regarding cigarette consumption, individuals with high food security reported a median daily intake of 11 cigarettes, while those with marginal, low, and very low food security consumed a median of 10, 20, and 15 cigarettes per day, respectively. Daily cigarette consumption varied by food security status, with higher consumption among those with lower food security with a level of significance 0.040.

Among the 819 participants, males outnumbered females (432 vs. 387). The majority of both males (55%) and females (45%) had high food security; however, this difference was not statistically significant ($p = 0.13$). Severe depression was more prevalent among individuals with very low food security (28%), compared to those with marginal (4.5%) and low food security (15%), demonstrating a significant association ($p < 0.001$). Similarly, severe anxiety was significantly higher among participants with very low food security (27%) compared to those with high food security (7%) ($p < 0.001$).

In terms of physical health conditions, 8.9% of participants with very low food security had cardiovascular disease ($p = 0.2$). The prevalence of hypertension was similar among those with high (87%) and very low food security (84%) ($p = 0.8$). Among participants with diabetes, the prevalence was equal (22%) for those with high and very low food security ($p = 0.5$). Additionally, 42% of participants with low food security were obese, though this association was not statistically significant ($p = 0.8$). The marital status of participants showed mixed results with p-value of $< 0.001$.

The plot (Fig 1) of No. of daily cigarette consumption indicates dispersion of data. By comparing the value of mean and variance it's verified, the count variable has overdispersion.

We have presented our model for analyzing the presence of two types of diseases, such as Cardiovascular and Diabetes, within the overall dataset. After filtering the data for the presence of cardiovascular disease, we found 114 observations, and 172 for diabetes. We have focused on reporting the most significant and common findings in both models since we are considering two diseases. Additionally, to compare how well CMP-GAM captures the over and under dispersion of data, we compared the model with Poisson-GAM and Negative Binomial-GAM.

**For Cardiovascular disease**

We've considered the following to model the CMP with the presence of cardiovascular disease.

$$
\begin{aligned}
ln(\lambda) = \beta_0 + \ & \beta_1\,(FDSCAT4\_A) + \beta_2\,(Sex) + \beta_3\,(HYPDIF\_A) \\
& + \beta_4\,(GADCAT\_A) + \beta_5\,(PHQCAT\_A) + s(BMI) + s(AGE\_A) \\
& + s(SMKAGE\_A) + s(SLPHRS\_A)
\end{aligned}
$$

$$
ln(\nu) = \gamma_o
$$

The coefficient significance results are described in Tables 2 and 3. The non-parametric components are Age, Smoking age, Sleep hours, BMI and the parametric terms are Anxiety and Depression scale, Hypertension, Food Security.

Looking at the results of the parametric components presented in Table 2, it is evident that food security, gender of participants, Hypertension, and the severity of Anxiety and depression have no significant association with daily cigarette

**Table 1. Demographic and clinical characteristics of adults in NIHS dataset with selected variables according to Food Security (Data Source: National Center for Health Statistics, National Health Interview Survey (NHIS), 2022).**

| Variable | Overall, N = 819[1] | High food security, N = 566 (69.1%) | Marginal food security, N = 89 (10.9%) | Low food security, N = 85 (10.4%) | Very low food security, N = 79 (9.6%) | p-value[1] |
|---|---|---|---|---|---|---|
| **Age, Median (IQR)** | 61 (51, 67) | 62 (54, 68) | 60 (49, 67) | 55 (44, 62) | 54 (43, 61) | **<0.001** |
| **Smoking Age, Median (IQR)** | 17 (15, 20) | 17 (15, 20) | 17 (15, 20) | 15 (13, 18) | 16 (14, 20) | **0.003** |
| **Daily Cigarette Consumption, Median (IQR)** | 12 (10,20) | 11 (10,20) | 10 (8,20) | 20 (10,20) | 15 (10,20) | **0.040** |
| **Sex, n (%)** | | | | | | 0.13 |
| Male | 432 (53%) | 309 (55%) | 46 (52%) | 45 (53%) | 32 (41%) | |
| Female | 387 (47%) | 257 (45%) | 43 (48%) | 40 (47%) | 47 (59%) | |
| **Depression Severity, n (%)** | | | | | | **<0.001** |
| None/minimal | 518 (63%) | 420 (74%) | 52 (58%) | 30 (35%) | 16 (20%) | |
| Mild | 163 (20%) | 89 (16%) | 23 (26%) | 29 (34%) | 22 (28%) | |
| Moderate | 74 (9.0%) | 32 (5.7%) | 10 (11%) | 13 (15%) | 19 (24%) | |
| Severe | 64 (7.8%) | 25 (4.4%) | 4 (4.5%) | 13 (15%) | 22 (28%) | |
| **Anxiety Severity, n (%)** | | | | | | **<0.001** |
| None/minimal | 564 (69%) | 447 (79%) | 60 (67%) | 33 (39%) | 24 (30%) | |
| Mild | 127 (16%) | 70 (12%) | 15 (17%) | 24 (28%) | 18 (23%) | |
| Moderate | 71 (8.7%) | 30 (5.3%) | 12 (13%) | 13 (15%) | 16 (20%) | |
| Severe | 57 (7.0%) | 19 (3.4%) | 2 (2.2%) | 15 (18%) | 21 (27%) | |
| **Cardiovascular disease, n (%)** | | | | | | 0.2 |
| Yes | 114 (14%) | 79 (14%) | 18 (20%) | 10 (12%) | 7 (8.9%) | |
| No | 705 (86%) | 487 (86%) | 71 (80%) | 75 (88%) | 72 (91%) | |
| **Cholesterol, n (%)** | | | | | | 0.8 |
| Yes | 453 (55%) | 313 (55%) | 52 (58%) | 48 (56%) | 40 (51%) | |
| No | 366 (45%) | 253 (45%) | 37 (42%) | 37 (44%) | 39 (49%) | |
| **Hypertension, n (%)** | | | | | | 0.5 |
| Yes | 703 (86%) | 491 (87%) | 72 (81%) | 74 (87%) | 66 (84%) | |
| No | 116 (14%) | 75 (13%) | 17 (19%) | 11 (13%) | 13 (16%) | |
| **Diabetics, n (%)** | | | | | | 0.8 |
| Yes | 172 (21%) | 123 (22%) | 15 (17%) | 17 (20%) | 17 (22%) | |

*(Continued)*

**Table 1.** (Continued)

| Variable | Overall, N = 819[1] | High food security, N = 566 (69.1%) | Marginal food security, N = 89 (10.9%) | Low food security, N = 85 (10.4%) | Very low food security, N = 79 (9.6%) | p-value[1] |
|---|---|---|---|---|---|---|
| No | 647 (79%) | 443 (78%) | 74 (83%) | 68 (80%) | 62 (78%) | |
| **BMI, n (%)** | | | | | | 0.7 |
| Underweight | 7 (0.9%) | 5 (0.9%) | 0 (0%) | 1 (1.2%) | 1 (1.3%) | |
| Healthy weight | 228 (28%) | 153 (27%) | 31 (35%) | 22 (26%) | 22 (28%) | |
| Overweight | 294 (36%) | 215 (38%) | 26 (29%) | 26 (31%) | 27 (34%) | |
| Obese | 290 (35%) | 193 (34%) | 32 (36%) | 36 (42%) | 29 (37%) | |
| **Marital Status, n (%)** | | | | | | <0.001 |
| Married but spouse is present | 232 (28%) | 192 (34%) | 19 (21%) | 12 (14%) | 9 (11%) | |
| Married but spouse is not present | 28 (3.4%) | 16 (2.8%) | 2 (2.2%) | 6 (7.1%) | 4 (5.1%) | |
| Widowed | 124 (15%) | 83 (15%) | 17 (19%) | 12 (14%) | 12 (15%) | |
| Divorced | 226 (28%) | 144 (25%) | 25 (28%) | 20 (24%) | 37 (47%) | |
| Separated | 22 (2.7%) | 15 (2.7%) | 3 (3.4%) | 2 (2.4%) | 2 (2.5%) | |
| Never Married | 126 (15%) | 81 (14%) | 15 (17%) | 17 (20%) | 13 (16%) | |
| Living with partner | 61 (7.4%) | 35 (6.2%) | 8 (9.0%) | 16 (19%) | 2 (2.5%) | |

[1] p-values measured from either Kruskal-Wallis rank sum test or Pearson's Chi-squared test.

consumption. The parametric components across all the subgroup analysis provides similar results, for this reason we've provided only the parametric values for cardiovascular disease.

From the non-parametric components of Table 3, smoking age shows significance in CMP-GAM. In Poisson-GAM, Age, smoking age, and sleep hours tend to have a significant effect on daily cigarette consumption. However, in Negative-GAM, no variable shows significance.

## For diabetics data

To assess diabetics data we've considered the following model with parameter,

$$l(\lambda) = \beta_o + \beta_1 (FDSCAT4\_A) + \beta_2 (HYPDIF\_A) + \beta_3 (GADCAT\_A)$$
$$+ \beta_4 (PHQCAT\_A) + \beta_5 (Sex) + \beta_6 (CHLEV\_A) + s(BMI)$$
$$+ s(AGE\_A) + s(SMKAGE\_A) + s(SLPHRS\_A)$$

$$ln(\nu) = \gamma_o$$

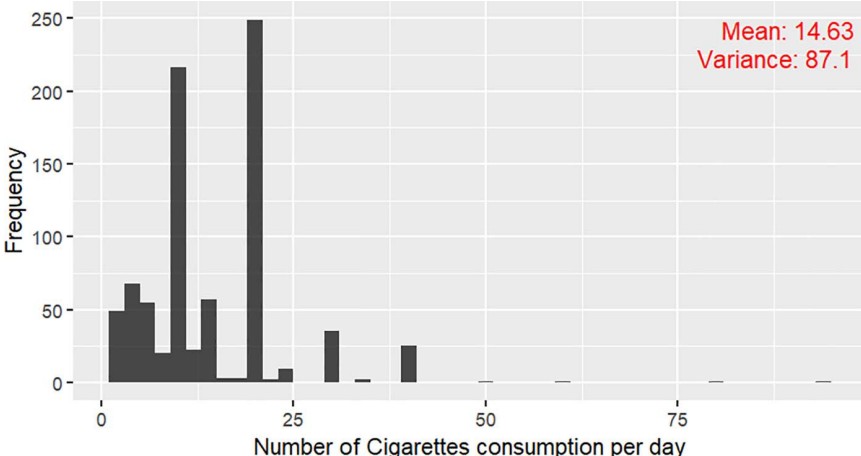

**Fig 1. Histogram of Daily Cigarette Consumption (Data Source: National Health Interview Survey (NHIS), 2022).**

The coefficient significance results are described in Table 4. The non-parametric components include Age, Smoking age, Sleep hours, and BMI, while the parametric terms consist of Food security, sex, Hypertension, Anxiety and Depression scale, and Cholesterol.

Like the cardiovascular dataset, no parametric variable was found to have significance in the diabetics model.

For overall data

We've considered the following model for overall dataset:

**Table 2. Comparison of parametric components of three models in the presence of Cardiovascular disease.**

|  | CMP-GAM (estimated) | Poisson-GAM (estimated) | NB-GAM (estimated) |
|---|---|---|---|
| **Parametric components** | | | |
| **Food Security** | | | |
| Marginal food security | −0.042 | −0.020 | −0.17 |
| Low food security | 0.048 | 0.34 | 0.19 |
| Very low food security | 0.06 | 0.015 | 0.24 |
| **Sex** | | | |
| Female | −0.03 | −0.069 | −0.12 |
| **Anxiety severity** | | | |
| Mild | 0.034 | 0.048 | 0.16 |
| Moderate | −0.066 | −0.24 | −0.34 |
| Severe | 0.025 | 0.28 | 0.06 |
| **Hypertension** | | | |
| No | −0.057 | −0.25 | −0.29 |
| **Depression severity** | | | |
| Mild | 0.015 | 0.13 | −0.04 |
| Moderate | 0.071 | 0.19 | 0.27 |
| Severe | 0.078 | 0.32 | 0.43 |

**Table 3. Comparison of non-parametric components of three models in the presence of Cardiovascular disease.**

|  | CMP-GAM (edf) | Poisson-GAM (edf) | NB-GAM (edf) |
|---|---|---|---|
| **Non-Parametric components** |  |  |  |
| s(BMI) | 1.00 | 1.001 | 1.00 |
| s(Age) | 1.00 | 6.839* | 1.00 |
| s(Smoking Age) | 2.87* | 7.686*** | 1.726 |
| s(Sleep hours) | 1.00 | 8.735*** | 1.00 |
| **AIC** | 828.04 | 1125.4 | 956.3 |

***p<0.001, **p<0.01, *p<0.05.

**Table 4. Comparison of non-parametric components of three models in presence of Diabetics.**

|  | CMP-GAM (edf) | Poisson-GAM (edf) | NB-GAM (edf) |
|---|---|---|---|
| **Non-Parametric components** |  |  |  |
| s(BMI) | 1.00 | 6.037*** | 1.00 |
| s(Age) | 7.64 | 8.67*** | 2.689* |
| s(Smoking Age) | 3.401* | 8.042*** | 2.790 |
| s(Sleep hours) | 1.93 | 8.312*** | 2.131 |
| **AIC** | 828.04 | 987.1 | 917.4 |

***p<0.001, **p<0.01, *p<0.05.

**Table 5. Comparison of non-parametric components in three models (overall data).**

|  | CMP-GAM (edf) | Poisson-GAM (edf) | NB-GAM (edf) |
|---|---|---|---|
| **Non-Parametric components** |  |  |  |
| s(BMI) | 2.78 | 4.378*** | 1.00 |
| s(Age) | 1.59 | 7.668*** | 1.663 |
| s(Smoking Age) | 4.02*** | 8.695*** | 2.262*** |
| s(Sleep hours) | 1.00 | 8.894*** | 1.003 |
| **AIC** | 5722.6 | 6000.2 | 5998.3 |

***p<0.001, **p<0.01, *p<0.05.

$$In(\lambda) = \beta_0 + \beta_1\,(FDSCAT4\_A) + \beta_2\,(HYPDIF\_A) \\ + \beta_3\,(GADCAT\_A) + \beta_4\,(PHQCAT\_A) + \beta_5\,(CHDEV\_A) \\ + \beta_6\,(DIBEV\_A) + \beta_7\,(CHLEV\_A) \\ + s(BMI) + s(AGE\_A) + s(SMKAGE\_A) + s(SLPHRS\_A)$$

$$In(\nu) = \gamma_o$$

The coefficient significance results are described in Table 5. The non-parametric components are Age, Smoking age, Sleep hours, and BMI, and the parametric terms are Food security, sex, Hypertension, Anxiety and Depression scale, Cardiovascular disease, Diabetic disease, and Cholesterol.

None of the parametric components yield significant results in any of the three models. Conversely, in the non-parametric components, smoking age is significant in CMP-GAM. For both the Poisson GAM and Negative Binomial GAM models, we obtain results similar to those of the diabetic model, only differing in the numeric value of the non-parametric coefficients.

Out of the three models, the Poisson GAM yields significant results for almost all smooth variables. On the other hand, only the age of smoking is a significant variable in CMP-GAM and NBGAM, supporting the reference by Chatla and Shmueli [22] that the Poisson GAM has more inference error than the other two models due to excessive dispersion.

Considering the overall dataset with an almost equal distribution of parametric and non-parametric components, CMP GAM and NB GAM are better with less inference error, despite our data only exhibiting overdispersion. Although we were unable to check for overdispersion, this could be useful for further research in the field of health science.

## 4 Discussion

This study used National Health Interview Survey 2022 data to understand the association between the daily consumption of cigarettes and household food security status among adults in the US. Our study revealed that the median daily consumption of cigarettes was higher for households with low to very low food security status. However, this study explored that for the overall population, households' food security status was not statistically significantly associated with daily cigarette consumption. Moreover, this association was unchanged for the two sub-groups such as cardiovascular and diabetic patients. Yet, research in the past has discovered that this connection can go both ways. For example, smoking could worsen food insecurity by using up financial resources, while food insecurity could cause psychological stress that encourages smoking and makes it harder to quit [61]. Our study was the first to use the CMP-GAM model to disentangle the association of daily cigarettes consumptions with household food security status.

This study observed a significant association between the age at which people started smoking and the number of cigarettes smoked per day, which is also prevalent in findings from Poisson-GAM and negative binomial-GAM that support some of the prior research findings. Individuals who initiated smoking at an early age tend to smoke more cigarettes daily [62]. Furthermore, beginning to smoke at a young age has been recognized as a strong indicator of nicotine dependence, suggesting that starting smoking early is linked to an increased probability of developing an addiction to nicotine and smoking a greater number of cigarettes each day [63]. At the same time, the prevalence of smoking has been found to be higher among food-insecure households compared to food-secure households.[5] This finding sheds additional light on the barely uncovered association between household food security status and smoking frequency among young adults.

Following the findings from sub-group analysis with diabetic's patients, results of Poisson-GAM and Negative-binomial-GAM observed an association between the age of the respondents and household food security status among patients with both cardiovascular and diabetics disease. Based on an earlier study it was found that the age of household heads was inversely related to household food security status [64]. Another study conducted in Eastern Cape Province, South Africa, found that age, household income, access to credit, and gender have a statistically significant impact on the food security status of households [65]. Additional research may be necessary to determine whether the age of the household head could have a direct impact on the household's food security status, especially for the households with diabetics patients.

Food security may also have an impact on the increased risk of depression and anxiety. Almost half of the participants in the study experiencing moderate to severe anxiety (50.8%) and depression (48.6%) also report low to very low household food security. Earlier evidence consistently indicates that food insecurity is significantly related to increased risks of depression, anxiety, and stress. The mental health impacts of food insecurity vary by region, with North American

households experiencing increased stress and stress [66]. Based on earlier research on Canadian adults, it was observed that food insecurity had been linked to adverse mental health outcomes for adults [67].

Higher cigarette consumption can be one of the crucial risk factors in the development and management of diabetes. One study showed that men who smoked 25 or more cigarettes per day had nearly double the risk of developing type 2 diabetes compared to non-smokers [68]. However, with the sub-group analysis of diabetic patients, this study did not find any significant relationship between the number of cigarettes smoked by respondents in a day and their household food security status. Likewise, this study found that the number of cigarettes smoked is not significantly associated with demographic variables like gender, BMI, sleep hours, presence of hypertension, or socioeconomic status. Aligned with this, earlier research with university students in Montenegro did not find a significant impact on cigarette smoking [69]. However, studies have found that BMI can be a factor that may influence smoking behavior. [70,71] Exploring this linkage further can be beneficial in planning and designing public health intervention aimed at reducing cigarette consumption.

### Limitations

This study has several limitations. The information is self-reported, which means there could be some loss of information due to potential biases related to recall and social desirability. The NHIS is a cross-sectional study and cannot track changes in health status or behaviors over time. Thus, it is essential to conduct repeat cross-sectional data or longitudinal research to determine the prevalence of cigarette smoking in this population. Besides this, our study was limited to assessing the association in the presence of only two types of diseases; exploring further with other diseases and considering the effects of comorbidities could have been guided into better construct in understanding the association. Additionally, this study did not conduct confounding or mediation analyses, which could have provided a deeper understanding of the pathways through which the exposure influences the outcome. Future research should consider incorporating these analyses to assess these relationships more comprehensively.

## 5 Conclusion

The study did not find any significant relationship between food security status and daily cigarette consumption both for overall and sub-categories of populations. However, it identified a significant association between daily cigarette consumption and the age of smoking initiation. This underscores the need for strengthening targeted intervention aims at preventing early smoking while promoting programs that educate young people about the risks of smoking and raising awareness of healthier lifestyle choices and risks of early smoking.

### Author contributions

**Conceptualization:** Shafeel Umam, Azizur Rahman.

**Formal analysis:** Shafeel Umam, Munniara Yesmin Munni.

**Methodology:** Shafeel Umam, Rubaiya Binte Razzak, Munniara Yesmin Munni.

**Project administration:** Azizur Rahman.

**Supervision:** Azizur Rahman.

**Writing – original draft:** Shafeel Umam, Rubaiya Binte Razzak, Munniara Yesmin Munni.

**Writing – review & editing:** Shafeel Umam, Rubaiya Binte Razzak, Azizur Rahman.

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
