## [Decision Letter · Decision Letter 0]

Dear Dr. Umam,

Thank you for submitting your manuscript to PLOS ONE. After careful consideration, we feel that it has merit but does not fully meet PLOS ONE’s publication criteria as it currently stands. Therefore, we invite you to submit a revised version of the manuscript that addresses the points raised during the review process.

This study focus on the association between food insecurity and daily cigarette consumption. Food insecurity and cigarette smoking are significant public health issues that disproportionately affect vulnerable populations. This study could provide some suggestions to deal with these problems.   

My main suggestion is that the authors should further clarify their hypothesis of the study and advance the data mining. Some points are listed as follows:

1. although this study used the data from NHIS, please also clarify the sampling method and the inclusion and exclusion criteria for the analyses;

2. please report the ethic statement for the study which the data collected from.

After a careful evaluation of your manuscript, the major revision should be made for the publication in PLOS ONE.

We look forward to receiving your revised manuscript.

Kind regards,

Ziqian Zeng

Academic Editor

PLOS ONE

2. Please provide captions for Table 5 in your manuscript.

3. Thank you for uploading your study's underlying data set. Unfortunately, the repository you have noted in your Data Availability statement does not qualify as an acceptable data repository according to PLOS's standards.At this time, please upload the minimal data set necessary to replicate your study's findings to a stable, public repository (such as figshare or Dryad) and provide us with the relevant URLs, DOIs, or accession numbers that may be used to access these data. For a list of recommended repositories and additional information on PLOS standards for data deposition, please see https://journals.plos.org/plosone/s/recommended-repositories .

Additional Editor Comments:

This study focus on the association between food insecurity and daily cigarette consumption. Food insecurity and cigarette smoking are significant public health issues that disproportionately affect vulnerable populations. This study could provide some suggestions to deal with these problems.

My main suggestion is that the authors should further clarify their hypothesis of the study and advance the data mining. Some points are listed as follows:

1.although this study used the data from NHIS, please also clarify the sampling method and the inclusion and exclusion criteria for the analyses;

2.please report the ethic statement for the study which the data collected from.

After a careful evaluation of your manuscript, the major revision should be made for the publication in PLOS ONE.

Reviewers' comments:

Reviewer's Responses to Questions

**Comments to the Author**

1. Is the manuscript technically sound, and do the data support the conclusions?

Reviewer #1: Yes

Reviewer #2: Yes

Reviewer #3: Partly

2. Has the statistical analysis been performed appropriately and rigorously?

Reviewer #1: I Don't Know

Reviewer #2: Yes

Reviewer #3: Yes

3. Have the authors made all data underlying the findings in their manuscript fully available?

Reviewer #1: Yes

Reviewer #2: Yes

Reviewer #3: Yes

4. Is the manuscript presented in an intelligible fashion and written in standard English?

Reviewer #1: Yes

Reviewer #2: Yes

Reviewer #3: No

Reviewer #1: This valuable manuscript investigates the relationship between food security and the number of daily smokers, addressing a critical public health concern. The use of the COM-Poisson Generalized Additive Model (CMP-GAM) to capture non-linear relationships in count data is both innovative and appropriate for the research question.

I believe this is a well-conducted study. However, I have several major comments that I hope will help the authors improve the manuscript further.

Reviewer #2: 1. The inclusion of too much nonlinearity will reduce the interpretability of the model. Does the author have any countermeasures for this?

2. This is a cross-sectional data. In addition to subgroup analysis, are other bias control measures taken?

Reviewer #3: The authors investigated the non-linear association between food security and daily cigarette consumption, they used a novel statistic approach, COM-Poisson generalized additive model, the results indicated food security status was not significantly affect the daily cigarette consumption, the age of smoking initiation was strong leading the daily cigarette consumption. The results seem nothing new and updated.

Aim to increase the transparence and understandable of the results, here are suggestions for your reference:

1. line 49-71, the background of reason of food insecurity to increase or decrease were not enough to support your hypothesis.

2. line 72-85, some sentences need to be segment, they are quite difficult to understand. The writing structure seems like evidences stacks, however, the reason of non-linear were not introduced were not present well.

3. line 176, 2.1.1 sub-group analysis, do you put the appropriate place?

4. Table 1, why not added the statistic, such as t, chi2, df and p-value?

5. Figure 1, the unit missed, difficult to see clear.

**Do you want your identity to be public for this peer review?** For information about this choice, including consent withdrawal, please see our Privacy Policy

Reviewer #1: No

Reviewer #2: No

Reviewer #3: **Yes: ** Fan Xu

---

## [Author Response · Author response to Decision Letter 1]

28 Mar 2025

Additional Editor

The responses to specific comments are as below

1. Although this study used the data from NHIS, please also clarify the sampling method and the inclusion and exclusion criteria for the analyses

Response

Thank you for your insightful comment. We have incorporated details on both the sampling method and the inclusion and exclusion criteria for the survey (in lines 140-158), as well as the inclusion and exclusion criteria for our analysis (in lines 166-171). These updates have been added to the Methods section for clarity. We appreciate your feedback in strengthening the transparency of our study.

Sampling method and the inclusion and exclusion criteria for the survey

The NHIS is an annual household survey representing the civilian non-institutionalized US population across the 50 states and the District of Columbia. It collects data from individuals residing in households and noninstitutional group quarters, including homeless shelters, rooming houses, and group homes. Individuals temporarily living in student dormitories or temporary housing are sampled within their permanent households.(1) The survey excludes those without a fixed household address, active-duty military personnel and civilians on military bases, residents of long-term care institutions (such as nursing homes and chronic care hospitals), individuals in correctional facilities, and U.S. nationals living abroad.(1) However, civilians residing with military personnel in non-military housing remain eligible for inclusion. The NHIS employs a geographically clustered sampling design to select dwelling units, ensuring national representativeness.(1) Based on the 2010 decennial census, the sampling process partitions the United States into 1,689 geographic areas, comprising counties, county equivalents, or contiguous county groups, ensuring they remain within state boundaries.(1) The sample is structured in a way that monthly data is nationally representative.(1) Usually, interviews for the NHIS are conducted in-person modules and followed up via telephone. For over 50 years, the U.S. Census Bureau has sent interviewers to American homes to inquire about various health topics. Data collection is continuous, spanning from January to December each year, allowing for timely and comprehensive health surveillance.(1)

Inclusion and exclusion criteria for the analysis

The inclusion and exclusion criteria for this study were determined based on the variables of interest and their alignment with the research objectives. From the 27,651 responses, (n=1,267) data were excluded as missing food security information. Additionally, non-smokers were excluded from the study (n=25,565). Out of 27,651 respondents, only 819 met the inclusion criteria based on the specified variables. The final selection was guided by the study's research objectives to ensure relevance and accuracy in the analysis.

2. please report the ethic statement for the study which the data collected from.

Response

Thank you for your valuable suggestion. We have now included the ethics statement for the study, specifying the ethical considerations related to data collection. This update has been added to the Methods section in lines 369-376 for clarity. We appreciate your feedback in enhancing the completeness of our manuscript.

Ethics statement

The Research Ethics Review Board (ERB) of the National Center for Health Statistics (NCHS) approved the content and methods of the National Health Interview Survey (NHIS) to protect the study participants.(2) Prior to taking part in the NHIS study, all participants provided their verbal consent. Potential NHIS respondents were informed about their rights regarding survey participation and were assured of the confidentiality of their responses, as well as the protection of their data.(3) All data in the publicly available dataset are fully anonymized prior to release. All authors declare they have no competing interests.

Reviewer- 2

3. The inclusion of too much nonlinearity will reduce the interpretability of the model. Does the author have any countermeasures for this?

Response

We appreciate the reviewer’s concern regarding model interpretability in the presence of excessive nonlinearity. To address this, we have carefully balanced linear and nonlinear components in our CMP GAM model. Specifically, we included a linear term in the model to account for the overall trend while allowing nonlinearity to capture subtle deviations where appropriate.

4. This is a cross-sectional data. In addition to subgroup analysis, are other bias control measures taken?

Response

We highly appreciate the reviewer’s thoughtful feedback regarding bias management in a cross-sectional study. While our current analysis did not explicitly account for confounders or mediation effects, we acknowledge their potential influence on our findings. In this study, we primarily focused on subgroup analysis to explore variations in associations. However, we recognize the importance of addressing confounding and mediation to better understand the pathways through which the exposure influences the outcome, and we mentioned this accordingly in the limitations section of the paper in lines (545 – 548). As part of our future research, we will incorporate causal mediation analysis and confounder adjustment techniques, such as stratification and regression-based controls, to assess these relationships comprehensively. We sincerely appreciate this valuable feedback and will carefully consider these aspects in subsequent work.

Added details to the limitations section

“Additionally, this study did not conduct confounding or mediation analyses, which could have provided a deeper understanding of the pathways through which the exposure influences the outcome. Future research should consider incorporating these analyses to assess these relationships more comprehensively.”

Reviewer- 3

5. Line 49-71, the background of reason of food insecurity to increase or decrease were not enough to support your hypothesis.

Response

Thank you for your valuable suggestions. We have incorporated the supporting literature reviews and evidence into the specified section. The relevant updates can be found between line numbers 49 and 76 and mentioned below.

“Previous research has shown that socioeconomic factors, including food security, are closely linked to smoking behavior. An earlier study among young adults observed that smoking prevalence is significantly higher for young adults who reported being food insecure.(4) The effect of food insecurity on smoking behavior is particularly noticeable among low-income and marginalized populations. Low-income adults experiencing food insecurity have been found to have a disproportionately high prevalence of cigarette smoking.(5) The impact of food insecurity on smoking behavior is multifaceted. Studies have shown that food insecurity is a barrier to smoking cessation.(6) Previous studies have discovered that facing food insecurity is associated with a higher likelihood of starting smoking for those who do not smoke.(7) Furthermore, Baek et al. noted that shifts from food security to insecurity were associated with almost a twofold rise in the likelihood of smoking behavior.(8) Individuals experiencing food insecurity may use smoking as a coping mechanism to manage stress and anxiety associated with their precarious food situation.(9) Canales et al. (2015) delineated that worrying about running out of food is an important indicator of food security and has direct causal implications on one’s perceptions of stress or anxiety. Whereas cigarette smoking is a common way to relieve tension and stress,(10) suggesting an association between the intensity of smoking and food insecurity. Individuals experiencing food insecurity may turn to smoking as a way to manage stress or suppress appetite, which can further complicate efforts to quit smoking.(11) Additionally, financial constraints can lead to prioritizing immediate relief through smoking over long-term health investments.(12) Psychological distress exacerbated by food insecurity may promote smoking for stress relief, creating a cycle that hinders cessation.(4) Moreover, a dose-response relationship has been observed between tobacco product use and levels of food insecurity, indicating that higher levels of smoking-induced financial deprivation can worsen food insecurity.(13) Nicotine intake reduces appetite and suppresses hunger,(14) which increases the likelihood of smokers using smoking as a coping mechanism to reduce hunger related to food scarcity. This dynamic exacerbates health disparities and significantly burdens public health systems. Taking these into account, there is a growing recognition of the need for more comprehensive assessments to understand the dynamics between tobacco use and food insecurity.”

6. line 72-85, some sentences need to be segment, they are quite difficult to understand. The writing structure seems like evidences stacks, however, the reason of non-linear were not introduced were not present well.

Response

We appreciate your insightful feedback. We have revised the sentences in lines 77-89 to improve clarity and readability. The writing structure has been adjusted, and a new section has been added to better introduce and present the reasons for the non-linear approach in lines 90 and 104. We hope these changes address your concerns and enhance the overall coherence of the section.

Revised paragraph

Numerous statistical models have been utilized in research studies to explore the association between food insecurity and smoking behaviors. In most of the research studies, logistic regression analysis was used to examine how food insecurity is associated with smoking status. (6,10,15) Longitudinal logistic and Tobit regressions analyzed the link between food security and smoking behaviors in low-income women, focusing on the longitudinal impact of food insecurity on smoking habits.(16) A prior study used multinomial logistic regression to assess and characterize the relationship between food security and tobacco product use.(17) Another study utilized multinomial logistic regression to examine the association between transitions of food insecurity with smoking cessation and shifts in the consumption pattern over two years among adults.(18) However, most studies only considered a linear association between food security and cigarette smoking behavior. Nevertheless, the non-linear nature of the association requires further investigation. To our knowledge, there is no study that has been conducted to capture the non-linear association between those variables using an appropriate count data model.

Reasons for incorporating non-linearity

Considering a non-linear association between food security and cigarette smoking behavior is important for several reasons:

Complex Relationships: The relationship between food security and smoking behavior may not be straightforward. Non-linear models can capture more complex interactions and thresholds that linear models might miss. For example, the impact of food insecurity on smoking behavior might be more pronounced at certain levels of food insecurity.(4)

Bidirectional Influences: Food insecurity and smoking behavior can influence each other in multiple ways. For instance, food insecurity might lead to increased stress and smoking as a coping mechanism, while smoking can exacerbate financial strain, worsening food insecurity.(10)

Policy and Intervention Design: Understanding non-linear relationships can help in designing more effective policies and interventions. For example, interventions might need to be targeted differently at various levels of food insecurity to be effective.

Literature Contribution: Incorporating non-linear associations enriches the literature by providing a more nuanced understanding of the dynamics between food security and smoking. It challenges the oversimplified view of linear relationships and encourages further research into the underlying mechanisms.

Overall, by considering non-linear associations, researchers can uncover more detailed insights and develop more targeted strategies to address the intertwined issues of food insecurity and smoking behavior.

7. line 176, 2.1.1 sub-group analysis, do you put the appropriate place?

Response

We appreciate your concern regarding the placement of Section 2.1.4 (Sub-group Analysis). After reviewing the structure, we confirm that this section is appropriately positioned within the manuscript. However, we have made minor adjustments to the arrangement to improve clarity and logical flow. We hope this revision enhances readability and aligns better with the overall structure of the paper.

8. Table 1, why not added the statistic, such as t, chi2, df and p-value?

Response

Thank you for your valuable insight. We have now included chi-square test statistics, including p-values in Table 1. Additionally, the interpretation of the chi-square results has been incorporated into the updated manuscript from lines 394–413 (as follows).

“The median age of the sample population was 61 years, with smoking initiation occurring at a median age of 17 years (p < 0.001 and p = 0.003, respectively). Regarding cigarette consumption, individuals with high food security reported a median daily intake of 11 cigarettes, while those with marginal, low, and very low food security consumed a median of 10, 20, and 15 cigarettes per day, respectively. Daily cigarette consumption varied by food security status, with higher consumption among those with lower food security with a level of significance 0.040.

Among the 819 participants, males outnumbered females (432 vs. 387). The majority of both males (55%) and females (45%) had high food security; however, this difference was not statistically significant (p = 0.13). Severe depression was more prevalent among individuals with very low food security (28%), compared to those with marginal (4.5%) and low food security (15%), demonstrating a significant association (p < 0.001). Similarly, severe anxiety was significantly higher among participants with very low food security (27%) compared to those with high food security (7%) (p < 0.001).

In terms of physical health conditions, 8.9% of participants with very low food security had cardiovascular disease (p = 0.2). The prevalence of hypertension was similar among those with high (87%) and very low food security (84%) (p = 0.8). Among participants with diabetes, the prevalence was equal (22%) for those with high and very low food security (p = 0.5). Additionally, 42% of participants with low food security were obese, though this association was not statistically significant (p = 0.8). The marital status of participants showed mixed results with p-value of <0.001.”

9. Figure 1, the unit missed, difficult to see clear.

Response

Thank you for pointing this out. The histogram represents the number of cigarettes consumed daily. We have updated Figure 1 by adding the appropriate unit (number of cigarettes consumed in a day) to enhance clarity. The revised histogram has been incorporated into the manuscript.

---

## [Decision Letter · Decision Letter 1]

Dear Dr. Umam,

Thank you for submitting your manuscript to PLOS ONE. After careful consideration, we feel that it has merit but does not fully meet PLOS ONE’s publication criteria as it currently stands. Therefore, we invite you to submit a revised version of the manuscript that addresses the points raised during the review process.

We look forward to receiving your revised manuscript.

Kind regards,

Ziqian Zeng

Academic Editor

PLOS ONE

Journal Requirements:

Additional Editor Comments :

The authors have revised manuscript according to the comments of reviewers. However, the minor revision is still needed.

Reviewers' comments:

Reviewer's Responses to Questions

**Comments to the Author**

Reviewer #1: All comments have been addressed

Reviewer #2: All comments have been addressed

Reviewer #3: All comments have been addressed

2. Is the manuscript technically sound, and do the data support the conclusions?

Reviewer #1: Yes

Reviewer #2: Yes

Reviewer #3: Yes

3. Has the statistical analysis been performed appropriately and rigorously?

Reviewer #1: Yes

Reviewer #2: Yes

Reviewer #3: Yes

4. Have the authors made all data underlying the findings in their manuscript fully available?

Reviewer #1: Yes

Reviewer #2: Yes

Reviewer #3: Yes

5. Is the manuscript presented in an intelligible fashion and written in standard English?

Reviewer #1: Yes

Reviewer #2: Yes

Reviewer #3: Yes

Reviewer #1: The authors addressed my comments; however, I have two additional minor comments.

1. Report Median and IQR: For the variable "Daily Cigarette Consumption," include both the median and the interquartile range (IQR) if second column of Table 1. This provides a clearer picture of the central tendency and variability.

2. Specify Indicators Clearly: For each variable, explicitly state which statistical indicators are being reported in front of the variable name in table 1. For example, for Daily Cigarette consumption you can use: Daily Cigarette consumption (Median, IQR).

Reviewer #2: (No Response)

Reviewer #3: No comment at current stage. All points has been revised accordingly, i am satisfied and agreed with author's revision.

**Do you want your identity to be public for this peer review?** For information about this choice, including consent withdrawal, please see our Privacy Policy

Reviewer #1: No

Reviewer #2: No

Reviewer #3: **Yes: ** Fan Xu

---

## [Author Response · Author response to Decision Letter 2]

10 May 2025

Dear Editor-in-Chief,

We, the authors of PONE-D-24-42111R1 titled “Exploring the non-linear association of daily cigarette consumption behavior and food security- An application of CMP GAM Regression,” appreciate the reviewer’s comments, who managed the time to examine this manuscript thoroughly and provided us with valuable insights. These suggestions greatly aided us in enhancing the manuscript, which has now been updated. We are confident that the content and clarity are significantly better in the revised version.

Response to the journal requirements are as follows:

Journal Requirements:

Response

Thank you for your guidance regarding the reference list. We have carefully reviewed all cited references using both Zotero and EndNote citation management tools to identify any retracted articles. Based on this review, we confirm that there are no retracted articles included in our reference list.

Below are the responses to the reviewers’ comments. The source file (.docx) of the revised manuscript was attached in two forms: ‘with changes marked’ and our ‘revised manuscript.’

Reviewer- 1

Responses to the comments are provided below:

1. Report Median and IQR: For the variable "Daily Cigarette Consumption," include both the median and the interquartile range (IQR) if second column of Table 1. This provides a clearer picture of the central tendency and variability.

Response

Thank you for your helpful suggestion. We have included the median and the interquartile range (IQR) for “Daily Cigarette Consumption” in our manuscript. To enhance clarity, we have now also included this information explicitly in a footnote to Table 1 to ensure it is easily visible to readers.

2. Specify Indicators Clearly: For each variable, explicitly state which statistical indicators are being reported in front of the variable name in table 1. For example, for Daily Cigarette consumption you can use: Daily Cigarette consumption (Median, IQR).

Response

Thank you for this constructive feedback. To enhance clarity and maintain a clean presentation, we have specified the statistical indicators (e.g., Median and IQR for continuous variables such as age, smoking duration, and daily cigarette consumption) in a dedicated footnote to Table 1. This approach ensures consistency in interpretation while avoiding overcrowding the variable labels. A separate footnote is also provided for categorical variables, indicating frequencies and percentages (n, %).

We sincerely hope that the revised manuscript matches the journal’s standards. We sincerely appreciate the reviewers' and editors' time and thoughtful feedback, which have helped improve our manuscript. We have carefully addressed all comments and made the necessary revisions. We have uploaded all the required documents as per journal's requirement. Thank you for your consideration, and we look forward to your response.

---

## [Editor Report · Decision Letter 2]

Dear Dr. Umam,

Thank you for submitting your manuscript to PLOS ONE. After careful consideration, we feel that it has merit but does not fully meet PLOS ONE’s publication criteria as it currently stands. Therefore, we invite you to submit a revised version of the manuscript that addresses the points raised during the review process.

Based on the reviewer’s comments and the publication criteria, the authors should revise the content according to the comment.

We look forward to receiving your revised manuscript.

Kind regards,

Ziqian Zeng

Academic Editor

PLOS ONE

Journal Requirements:

Additional Editor Comments:

According to the reviewer’s comments and the publication criteria, my main suggestion is that the authors should further revise the format of tables. Some points are listed as follows:

1.For each variable in table 1, explicitly state which statistical indicators are being reported with the variable name. For example, for Daily Cigarette consumption you can use: Daily Cigarette consumption (Median, IQR).

It is not suitable to make a note to describe in the table. The authors should revise the format according to the author’s guide.

---

## [Author Response · Author response to Decision Letter 3]

15 Jun 2025

Dear Editor-in-Chief,

We, the authors of PONE-D-24-42111R2 titled “Exploring the non-linear association of daily cigarette consumption behavior and food security- An application of CMP GAM Regression,” appreciate the reviewer’s comments, who managed the time to examine this manuscript thoroughly and provided us with valuable insights. These suggestions greatly aided us in enhancing the manuscript, which has now been updated. We are confident that the content and clarity are significantly better in the revised version.

Responses to the journal requirements are as follows:

Journal Requirements:

Response

Thank you for your guidance regarding the reference list. We have carefully reviewed all cited references using both Zotero and EndNote citation management tools to identify any retracted articles. Based on this review, we confirm that there are no retracted articles included in our reference list.

Response

Thank you for the instruction. We have uploaded our figure to the PACE digital diagnostic tool and reviewed the output to ensure it meets PLOS figure requirements. The revised figure, validated through PACE, has been incorporated into the updated manuscript accordingly. Please let us know if any additional adjustments are needed.

Below are the responses to the Additional Editor’s comments. The source file (.docx) of the revised manuscript was attached in two forms: ‘with changes marked’ and our ‘revised manuscript.’

Additional Editor

Responses to the comments are provided below:

1. For each variable in table 1, explicitly state which statistical indicators are being reported with the variable name. For example, for Daily Cigarette consumption you can use: Daily Cigarette consumption (Median, IQR).

It is not suitable to make a note to describe in the table. The authors should revise the format according to the author’s guide.

Response

Thank you for the helpful suggestion. We have revised Table 1 to explicitly include the statistical indicators (e.g., Median (IQR); n (%)) for each variable name, as recommended. In making these changes, we followed the formatting structure used in Table 1 (on page 5) of a previously submitted and accepted PLOS ONE article titled "Measure what matters: A survey-based examination of health equity tracking and measurement practices across healthcare systems in the United States."(1) We hope this revised format now adheres to the PLOS ONE author guidelines and reflects consistency with the journal's published standards.

We appreciate your guidance and hope this revision addresses the concern.

We sincerely hope that the revised manuscript matches the journal’s standards. We sincerely appreciate the reviewers' and editors' time and thoughtful feedback, which have helped improve our manuscript. We have carefully addressed all comments and made the necessary revisions. Thank you for your consideration, and we look forward to your response.

Best regards,

Shafeel Umam,

Corresponding Author,

Department of Statistics and Data Science,

Jahangirnagar University, Savar, Dhaka-1342

Email: shafeel.umam@outlook.com

Rubaiya Binte Razzak,

Department of Statistics and Data Science,

Jahangirnagar University, Savar, Dhaka-1342

Email: razzakrubaiya@gmail.com

Munniara Yesmin Munni,

PME department,

Christian Commission for Development in Bangladesh

Email: ymunniara@gmail.com

Azizur Rahman,

Department of Statistics and Data Science,

Jahangirnagar University, Savar, Dhaka-1342

Email: rahman.aziz83@gmail.com

Reference:

1. Roth H, Marchis ED, Kopaskie K, Restall A, Fichtenberg C, Ray S, et al. Measure what matters: A survey-based examination of health equity tracking and measurement practices across healthcare systems in the United States. PLOS ONE. 2025 May 21;20(5):e0323381.

---

## [Editor Report · Decision Letter 3]

Exploring the non-linear association of daily cigarette consumption behavior and food security- An application of CMP GAM Regression

PONE-D-24-42111R3

Dear Dr. Umam,

We’re pleased to inform you that your manuscript has been judged scientifically suitable for publication and will be formally accepted for publication once it meets all outstanding technical requirements.

Kind regards,

Ziqian Zeng

Academic Editor

PLOS ONE

Additional Editor Comments (optional):

The authors have addressed the concerns, please double-check the format of tables according to the three-line table setting. 
---

## [Editor Report · Acceptance letter]

PONE-D-24-42111R3

PLOS ONE

Dear Dr. Umam,

I'm pleased to inform you that your manuscript has been deemed suitable for publication in PLOS ONE. Congratulations! Your manuscript is now being handed over to our production team.

Kind regards,

on behalf of

Dr. Ziqian Zeng

Academic Editor

PLOS ONE